# Effect of Probiotic *Lactobacillus plantarum* Dad-13 on Metabolic Profiles and Gut Microbiota in Type 2 Diabetic Women: A Randomized Double-Blind Controlled Trial

**DOI:** 10.3390/microorganisms10091806

**Published:** 2022-09-09

**Authors:** Ninik Rustanti, Agnes Murdiati, Mohammad Juffrie, Endang Sutriswati Rahayu

**Affiliations:** 1Department of Food and Agricultural Product Technology, Faculty of Agricultural Technology, Universitas Gadjah Mada, Jl. Flora No 1 Bulaksumur, Yogyakarta 55281, Indonesia; 2Department of Nutrition Science, Faculty of Medicine, Universitas Diponegoro, Jl. Prof Soedarto, Tembalang Semarang 50275, Indonesia; 3Faculty of Medicine, Public Health and Nursing, Universitas Gadjah Mada, Jl. Farmako Sekip Utara, Yogyakarta 55281, Indonesia; 4Center for Food and Nutrition Studies, Universitas Gadjah Mada, Jl. Teknika Utara Barek, Yogyakarta 55281, Indonesia; 5Center of Excellence for Probiotics, Universitas Gadjah Mada, Yogyakarta 55281, Indonesia

**Keywords:** probiotic, anthropometric, SCFA, pH fecal, HbA1c

## Abstract

Several pathways link type 2 diabetes (T2D) mellitus to the gut microbiome. By modifying the gut microbiota (GM), probiotics may be useful in the treatment of T2D. *Lactobacillus plantarum* Dad-13 is an indigenous Indonesian probiotic strain that has colonized the digestive tracts of healthy Indonesian adults. Furthermore, the GM of Indonesians is dominated by *L. plantarum*. The probiotic *L. plantarum* Dad-13 is likely suitable for Indonesians. This study aimed to assess the effect of the probiotic *L. plantarum* Dad-13 on metabolic profiles and GM of women with T2D in Yogyakarta, Indonesia. Twenty women from each group of forty T2D patients received either a probiotic or a placebo. The probiotic group consumed 1 g skim milk powder containing 10^10^ CFU/g *L. plantarum* daily for 11 weeks. The placebo group received 1 g skim milk powder only daily for 11 weeks. At the start and end of the experiment, anthropometric measures, dietary intake surveys, blood samples, and fecal samples were obtained. The GM analysis of all samples was performed using polymerase chain reaction, and Illumina Novaseq was applied to the selected samples from each group at the beginning and end of the trial. Short-chain fatty acids (SCFAs) were analyzed with gas chromatography. The level of HbA1c in the probiotic group (n:10) significantly decreased from 9.34 ± 2.79% to 8.32 ± 2.04%. However, in comparison with the placebo (n:8), *L. plantarum* Dad-13 supplementation did not significantly decrease the HbA1c level. No significant change was observed in the fasting blood sugar and total cholesterol levels in either group. The GM analysis showed that *L. plantarum* Dad-13 supplementation resulted in a considerable increase in the *L. plantarum* number. No significant changes were observed in the *Bifidobacterium* and *Prevotella* populations. In addition, no significant change was observed in the fecal pH and SCFA (e.g., acetic acid, propionate, butyrate, and total SCFA) after supplementation with *L. plantarum* Dad-13.

## 1. Introduction

Type 2 diabetes mellitus (T2D) is a metabolic disease characterized by high blood glucose levels resulting from a deficiency in insulin synthesis, insulin resistance, or both. Diabetes and its effects have become a major global health problem. According to the International Diabetes Federation (IDF), around 463 million (9.7%) adult people 20–79 years of age had diabetes in 2019, with the number anticipated to rise to 700 million (10.9%) by 2045 [1].

Dysbiosis, or an imbalance in microbial homeostasis, is produced by alterations in the gut microbiota (GM) and contributes to the development of chronic diseases such as T2D. Dysbiosis of the GM affects glucose metabolism improvement significantly. Patients with T2D have altered intestinal microbiota, with a reduced Bacteroidetes Firmicutes ratio (F/B) and fewer bacteria (e.g., *Bifidobacteria*); however, increased levels were observed in Gram-negative bacteria that create endotoxins that disrupt the host’s metabolic function [2,3,4,5].

Probiotics may help patients with T2D management by modifying the GM. Probiotics are live bacteria that can alter the intestinal microbiota when taken as a food or supplement. *Lactobacillus* is the most common probiotic. According to meta-analysis research, treatment with probiotics can lower the HbA1c level, fasting blood glucose (FBS) level, and insulin resistance in T2D patients [6,7].

*Lactobacillus plantarum* Dad-13 is an indigenous Indonesian probiotic strain that has survived and colonized in the gastrointestinal tracts of healthy Indonesian adults [8,9]. *E. coli* and non-*E. coli* coliform bacteria were reduced in Indonesian school-aged children who consumed *L. plantarum* Dad-13 powder [10]. In the GM of the average Indonesian, *L. plantarum* is a dominating bacterium [11]. The GM composition changed after the treatment with gummy *L. plantarum* Dad-13. In addition, the treatment with *L. plantarum* Dad-13 helped moderately malnourished newborns improve their anthropometry and nutritional health [12]. The body weight and body mass index (BMI) in overweight adults reduced after ingesting the indigenous probiotic powder *L. plantarum* Dad-13 [13]. Therefore, *L. plantarum* is most likely a good probiotic for Indonesians. This study aimed to assess the influence of the powder of *L. plantarum* Dad-13 probiotic on the metabolic profiles and GM of women with T2D in Yogyakarta, Indonesia.

## 2. Materials and Methods

### 2.1. Study Design

The research was conducted in 2020 for intervention in public health centers in the Sleman regency, Yogyakarta, Indonesia. This 12-week study employed a randomized double-blind controlled trial research methodology. The COVID-19 pandemic’s restrictions forced a reduction in the intervention time to 11 weeks. For the 11-week study, the probiotic group consumed 1 g skim milk powder containing *Lactobacillus plantarum* Dad-13 10^10^ CFU each day, whereas the placebo group diet received 1 g skim milk powder daily.

Food intake, GM composition, and fecal short-chain fatty acids (SCFAs) were the primary outcomes of this study, whereas demographic data, anthropometric data, metabolic indicators, and physical activity were the supplementary results. At the study’s outset, demographic data, such as age, education, employment, income, T2D duration, medicines, and dose, were collected. At the beginning and end of the trial, anthropometric data, 7-day frequency of defecation, stool samples, and blood samples were obtained. The blood samples were used for metabolic indicators analyses, such as HbA1c, FBS, and cholesterol total analysis. By contrast, stool samples were used for the analyses of fecal characteristics (color, consistency, and pH), GM, and SCFAs. A semi-quantitative food frequency questionnaire and the International Physical Activity Questionnaire were used to gather information on food consumption and physical activity during the 4th and 11th weeks. The NutriSurvey 2007 program (http://www.nutrisurvey.de/ (accessed on 2 February 2020)) was used to measure the macronutrient consumption (energy, carbohydrate, protein, fat, fiber, and water).

The FBS, HbA1c, and total cholesterol levels were assessed at Yogyakarta’s Parahita Laboratory. The Biotechnology Laboratory and Waste Management Laboratory in the Faculty of Agricultural Technology, Universitas Gadjah Mada performed the DNA extraction and SCFA analysis. The polymerase chain reaction (PCR) study of the GM was performed at Universitas Gadjah Mada’s Laboratorium Penelitian dan Pengujian Terpadu, and the next-generation sequencing analysis was performed at Novogene Ltd. in Singapore.

### 2.2. Subject Participants

Subjects were obtained from 1146 women with T2D who visited 13 public health centers in Sleman regency, Yogyakarta, Indonesia. They were screened in accordance with the requirements for inclusion in this study. The inclusion criteria for the T2D patients were as follows: between 20 and 50 years of age, BMI < 30, HbA1c ≥ 6.5%, not pregnant and not breastfeeding, not menopausal, not smoking, not drinking alcohol, not consuming antidiabetic drugs, and not consuming other drugs. The probiotic and placebo groups were formed by random selection of the participants. Using the Excel 2016 formula, RANDBETWEEN (1;40), randomization was carried out by an independent enumerator. Subjects with scores from 1 to 20 were assigned to the probiotic group, and those with scores from 21 to 40 were assigned to the placebo group. Prior to the study’s conclusion, neither the subjects nor the researchers were aware of the product. The exclusion criteria were going through probiotic and/or antibiotic therapy within the treatment period, being pregnant, or withdrawing consent during the study.

### 2.3. Ethical Approval

As a condition of participation, all subjects had to provide a written informed consent. The Medical and Health Research Ethics Committee, Faculty of Medicine, Public Health, and Nursing, Universitas Gadjah Mada authorized the study protocol, which followed the principles of the 1975 Declaration of Helsinki (Protocol number: KE/FK/1356/EC/2019; Approval date: 18 November 2019).

### 2.4. Fecal Sample Collection

Before collecting the patient’s stool on day 28 (plus one day), each patient received a fecal kit box, and the procedure was explained. After being instructed to defecate, the volunteer’s fecal matter was placed in fecal tubes. As quickly as feasible, a sample was placed in a fecal box with ice bags and brought to the laboratory. A fecal sample was placed in a second fecal tube with 2 mL RNA (Sigma-Aldrich; R0901; Saint Louis, MO, USA). It was promptly stored at freezing temperature (−40 °C) before being utilized [14,15].

### 2.5. pH and SCFA Analyzes

A pH meter (pH meter; Spear Eutech, Singapore) was used to determine the fecal pH. Following calibration, the probe was immediately inserted into the fecal sample, and monitoring was performed until a stable reading was established. For SCFA analysis, 0.2 g feces were weighed in a 2 mL microtube and injected with sterile aquabidest water. After 20 min of ultrasonication, the fecal suspension was centrifuged for 10 min (14,000 rpm, 4 °C). The supernatant was centrifuged for a second cycle (14,000 rpm at 4 °C for 10 min). A gas chromatograph with a flame ionization detector and a capillary column (Crossbond polyethylene glycol, 30 m × 0.25 mm × 0.25 m) was used to analyze the final supernatant (Shimadzu, GC-2010 Plus, Kyoto, Japan). With nitrogen as the gas carrier, the temperatures for the sample injection and detection were 250 °C (flow rate: 38.7 mL/min; pressure: 100 kPa).

### 2.6. DNA Extraction

The extraction of DNA from the fecal sample kicked off the sequencing procedure. A previously modified bead-beating approach was used to extract DNA [15]. After being diluted with RNA (*w/v*) ten times, the fecal sample was rinsed with 1 mL phosphate-buffered saline. The fecal sample was then violently mixed for 60 s at 4000 rpm with a bead beater with 300 µL Tris-sodium dodecyl sulfate solution and 500 µL Tris-ethylenediaminetetraacetic acid (TE) buffer-saturated phenol (FastPrep-24^TM^, MP Biomedials, Santa Ana, CA, USA). The recovered supernatant was mixed with 400 µL phenol/chloroform/isoamyl alcohol (25:24:1; *v/v*) and aggressively mixed for 90 s at 4000 rpm with a bead beater (FastPrep-24^TM^, MP Biomedials, Santa Ana, CA, USA), followed by centrifugation at 13,000 rpm at 4 °C for 5 min with a bead beater (FastPrep-24^TM^, MP Biomedials, Santa Ana, CA, USA). Afterward, 25 µL 3 M sodium acetate (pH 5.2) was added to 250 µL supernatant, and the mixture was then incubated for 30 min on ice. A volume of 300 µL isopropanol was added, and the mixture was centrifuged at 13,000 rpm at 4 °C for 5 min. After being hand-agitated and washed with 500 µL cold 70% ethanol, the DNA pellet was centrifuged at 4 °C for 5 min at 13,000 rpm. The final DNA pellet was air dried, suspended in 20 µL TE buffer (pH 8.0), and kept at −30 °C until required [14].

### 2.7. Real-Time Quantitative PCR (qPCR) Analysis

The real-time qPCR technique was utilized for the microbiota analysis step, which also involved DNA dilution from the findings of DNA isolation, the creation of a PCR master mix, reading, the creation of standard curves, and determining the total number of bacteria [16]. The concentration of bacterial DNA was increased to 20 ng/µL for *L. plantarum*, *Bifidobacterium*, and *Prevotella*. The PCR master mix was prepared using a mixture of 5 µL Eva Green, 0.5 µL × 1000 nM forward primer and reverse primer, 1 µL sample DNA, and 3 µL nuclease-free water. Bio-Rad CFX-96 was used for real-time PCR. The analysis program Bio-Rad CFX Manager Software 3.0 was used to determine the outcomes of bacterial DNA quantification and multiplication. The calculation of the number of cells was carried out using a standard curve based on the DNA concentration of *L. plantarum* Dad-13. The total amounts of *L. plantarum*, *Bifidobacterium*, and *Prevotella* were revealed in the study. Table 1 displays the DNA base sequences of the employed primers.

### 2.8. Sequencing Data Processing

The forward primer CCTAYGGGRBGCASCAG and the reverse primer GGACTACNNGGGTATCTAAT were used to amplify the 16S rRNA gene’s V3 and V4 sections. PCR amplification of targeted regions was performed by using specific primers connecting with barcodes. The PCR products with proper size were selected by 2% agarose gel electrophoresis. The same amount of PCR products from each sample was pooled, end-repaired, A-tailed, and further ligated with Illumina adapters. Libraries were sequenced on a paired-end Illumina platform to generate 250 bp paired-end raw reads. The library was checked with Qubit and real-time PCR for quantification and bioanalyzer for size distribution detection. Quantified libraries were pooled and sequenced on Illumina platforms (The NEB-Next^®^ UltraTM DNA Library Prep Kit for Illumina, San Diego, CA, USA), according to effective library concentration and data amount required.

### 2.9. Bioinformatic Analysis

Based on their special barcodes, samples were assigned to paired-end reads, which were reduced by deleting the barcode and primer sequences. Paired-end reads were combined using FLASH (V1.2.7) (http://ccb.jhu.edu/software/FLASH/ (accessed on 4 February 2021)), and the splicing sequences were known as raw tags when at least a portion of the reads overlapped with the read produced from the opposing end of the identical DNA fragment [18].

To obtain the high-quality clean tags, we conducted quality filtering of the raw tags using precise filtering parameters and the QIIME (V1.7.0) quality-controlled methodology (http://qiime.org/scripts/split_libraries_fastq.html (accessed on 4 February 2021)) [19,20]. The tags were compared with the gold database, which can be found at SILVA138 database http://www.arb-silva.de/ (accessed on 4 February 2021), to find chimera sequences using the UCHIME Algorithm (UCHIME Algorithm, http://www.drive5.com/usearch/manual/uchime_algo.html (accessed on 4 February 2021)). The chimeric sequences were subsequently eliminated. The most successful tags were finally acquired [21].

We assessed the sequence with the Uparse software (Uparse v7.0.1001, http://drive5.com/uparse/ (accessed on 4 February 2021)) using all functional tags. OTUs were allocated to sequences that were 97% identical. For each OTU, a sample sequence was evaluated for further annotation [22]. The SILVA Data-SSUrRNA database (http://www.arb-silva.de/ (accessed on 4 February 2021)) was utilized to perform species annotation for each sample sequence using the Mothur program at each taxonomic rank: kingdom, phylum, class, order, family, genus, and species (threshold: 0.81) [23,24]. MUSCLE (version 3.8.31, available at http://www.drive5.com/muscle/ (accessed on 4 February 2021)) can efficiently compare a large number of sequences to determine the evolutionary relationship between all OTU representative sequences [25]. The OTU abundance data were normalized using a sequence number standard that corresponded to the sample with the fewest sequences. The complexity of biodiversity was analyzed using the alpha diversity indicators (Chao1, observed species, Simpson, and Shannon). Furthermore, species complexity was assessed using beta diversity indices (unweighted and weighted unifrac). All these tests were carried out using QIIME (V1.7.0).

### 2.10. Statistical Analysis

The SPSS 17 for windows was used for statistical analysis. Data are shown as the mean ± standard deviation (SD). If the data were normal, an independent *t*-test was used to compare the groups, and if the data were not normal, a non-parametric Mann–Whitney test was employed. Meanwhile, a paired *t*-test was used to examine the differences within groups (before–after intervention) if the data were normal, and the Wilcoxon test was applied if the data were not normal.

## 3. Results

### 3.1. Baseline Characteristics

In this trial, 40 T2D women were enrolled; 20 were placed in the probiotic group and 20 in the placebo group. However, for each group, only 18 participants completed the trial. Two subjects from the placebo group were excluded because they declined to participate and had stopped consuming medicine. Meanwhile, two subjects from the probiotic group were excluded because they were moving away from the area and declined to participate further. Figure 1 shows the participant flow in accordance with CONSORT. All subjects received more than 87.0% product with a comparable compliance rate (placebo: 92.28% vs. probiotics: 91.85%). The probiotics and placebo were well-tolerated and generally accepted by the participants. During the study period, the participants reported no adverse effects. However, at the end of this study, only 18 subjects (8 subjects of the placebo and 10 subjects of the probiotic group) were permitted to provide their blood samples due to the COVID-19 pandemic.

Table 2 presents the baseline characteristics, anthropometric, and metabolic profiles of the subjects. Age, anthropometric measurements, and metabolic profiles at baseline did not differ significantly across the groups.

### 3.2. Anthropometric and Blood Pressure before and after Intervention

Table 3 shows the anthropometric and blood pressure of both groups before and after the intervention. No significant change was observed in the weight, BMI, waist circumferences, hip circumferences, and systolic and diastolic pressures of placebo and probiotic groups.

### 3.3. Fecal Characteristics and Defecation Frequency

Table 4 shows the fecal characteristics, including the pH, color, and consistency, and defecation frequency in both groups before and after the probiotic intervention. Four scales were used to indicate the color (1: yellow, 2: brownish-yellow, 3: brown; 4: green). The fecal consistency was identified with the Bristol chart. Following the administration of probiotic *L. plantarum* Dad-13, the frequency of defecation increased in the probiotic group. Before and after the intervention, the pH in the placebo group was 6.21 ± 0.61 and 6.08 ± 0.42, respectively. The pH level in the probiotic group was 6.28 ± 0.39 before and 6.21 ± 0.37 after the intervention.

### 3.4. SCFAs

Table 5 displays the concentrations of SCFA in the probiotic and placebo groups. In addition, neither group’s SCFA levels changed significantly (*p* > 0.05) as a result of the intervention. Similarly, the SCFA content of T2D individuals was unaffected by the administration of *L. plantarum* Dad-13 in this investigation.

### 3.5. Dietary Intake and Physical Activity

Table 6 displays the pre- and postintervention nutritional consumption and physical activity of the two groups. Both groups’ consumption of macronutrients, fiber, and water did not change significantly after the intervention. The physical activity in both groups also showed no significant change. 

### 3.6. Specific Bacteria Analyzed by PCR

*L. plantarum*, *Bifidobacterium*, and *Prevotella* were selected as the bacteria of interest, and qPCR analysis was used to calculate their populations (Table 7). *L. plantarum* was quantified to assess its gastrointestinal tract resistance.

A significant increment in the number of *L. plantarum* was observed in the probiotic group, but a significant decrement was detected in the placebo group. However, no appreciable variations were noticed in the proportions of *Prevotella* and *Bifidobacterium* in either group.

### 3.7. Metabolic Profiles before and after the Intervention 

Table 8 displays the metabolic characteristics of both groups. In the probiotic group, the FBS level dropped considerably. However, the HbA1c changes did not differ substantially between groups compared with the placebo group. Meanwhile, the FBS and total cholesterol did not change after administration of the probiotic.

### 3.8. GM Taxonomics between Groups

Figure 2 shows the F/B and the taxonomy (top 10 relative abundance) of each group. In both categories, Firmicutes, Bacteroidetes, Actinobacteria, and Proteobacteria dominated. A small proportion of the phyla Verrucomicrobiota, Fusobacteriota, Spirochaetota, Acidobacteriota, Chloroflexi, and Nitrospirota appeared in both groups.

The placebo and probiotics have no statistically significant effect at the phylum, family, and genus levels. Concerning the two dominant phyla in the two groups, the relative abundance of Firmicutes decreased, and that of Bacteroidetes decreased dramatically in the probiotic group. Nevertheless, the F/B in the probiotic group increased, and that of the placebo group decreased after the intervention (Figure 2).

The most dominant families in both groups were Prevotellaceae, Bacteroidaceae (Phylum Bacteroidetes), Bifidobacteriaceae, Coriobacteriaceae (Phylum Actinobacteria), Succinivibrionaceae (Phylum Proteobacteria), Lachnospiracea, Ruminococcaceae, Selenomonadaceae, Lactobacillaceae, and Oscillospiraceae (Phylum Firmicutes).

The top 10 relatively abundant genera in both groups were *Prevotella*, *Bacteroides* (Phylum Bacteroidetes), *Bifidobacterium* (Phylum Actinobacteria), *Succinivibrio* (Phylum Proteobacteria), *Agathobacter*, *Megamonas*, *Blautia*, *Faecalibacterium*, *Collinsella*, and *Lactobacillus* (Phylum Firmicutes). In both groups after the intervention, the abundance of *Prevotella* decreased, whereas those of *Succinivibrio* and *Faecalibacterium* increased. However, the abundance levels of *Bacteroides*, *Bifidobacterium*, *Megamonas*, *Blautia*, and *Collinsella* increased in the probiotic group but decreased in the placebo group. Meanwhile, *Agathobacter* and *Lactobacillus* were reduced in the probiotic group but increased in abundance in the placebo group. Not all genera were altered in response to probiotic use, according to a heatmap showing the top 35 relative abundances at the genus level for the placebo and probiotic groups (Figure 3).

### 3.9. GM Diversity and Composition

As illustrated in Figure 4a–d, the alpha diversity reflects the richness of the GM, as reflected by the observed species, Chao1, Simpson, and Shannon indices.

When evaluated using the Wilcoxon and Tukey tests, all the indices had no significant changes in the alpha diversity (*p* > 0.05). Beta diversity also reflects the variations in the GM, and it is determined using weighted and unweighted unifrac (Figure 5a,b). The unweighted unifrac was derived using the OTU phylogenetic connection, whereas the weighted unifrac was generated using the OTU abundance. After 11 weeks of dosing, no discernible difference was observed between the probiotic and placebo groups in beta diversity measurements (Figure 4a,b). Thus, probiotic intervention did not affect the composition and relative abundance of gut microbial communities in both groups, as determined by the Wilcoxon and Tukey tests (*p* > 0.05).

### 3.10. Correlation of Metabolic Profiles with SCFA, GM (Genus Level), and Food Intake

Spearman’s correlation analysis was applied to 18 subjects (8 subjects in the placebo group and 10 subjects in the probiotic group) who completed data collection for their metabolic profiles, SCFA, GM (genus level), and food intake after intervention. Figure 6 shows the correlation of the two parameters using corrplot.

HbA1c had a negative association with the quantitative numbers of Bifidobacterium (r: −0.669, *p*: 0.002) and Prevotella quantitative number (r: −0.566, *p*: 0.014) and relative abundances of Succinivibrio (r: −0.491, *p*: 0.038), Faecalibacterium (r: −0.658, *p*: 0.003), and Collinsela (r: −0.640, *p*: 0.004). HbA1c had a slightly positive association with body weight (r: 0.406, *p*: 0.095) and systolic blood pressure (r: 0.443, *p*: 0.066) but a slightly negative link with the relative abundances of Bifidobacterium and Bacteroides (r: −0.425, *p*: 0.079; r: −0.427, *p*: 0.077, respectively). FBS revealed a small positive relationship with systolic blood pressure (r: 0.428, *p*: 0.076). The total cholesterol had a positive correlation with systolic blood pressure (r: 0.577, *p*: 0.012). Meanwhile, the amount of Bifidobacterium (r: −0.433, *p*: 0.073) and fiber consumption had a slightly negative connection (r: −0.464, *p*: 0.053). Systolic blood pressure had a favorable relationship with Megamonas (r: 0.515, *p*: 0.029) and water (r: 0.497, *p*: 0.036) but a negative relationship with carbohydrate and fiber intake (r: −0.534, *p*: 0.023; r: −0.569, *p*: 0.014). Systolic blood pressure had a slightly positive relationship with body weight (r: 0.432, *p*: 0.073), BMI (r: 0.419, *p*: 0.084), and Bacteroides (r: 0.450, *p*: 0.061) but a slightly negative relationship with the Bifidobacterium number (r: −0.464, *p*: 0.053). Diastolic pressure had a negative correlation with the total SCFA (r: −0.507, *p*: 0.032), whereas acetic acid and butyric acid had a slightly negative correlation (r: −0.438, *p*: 0.069; r: −0.451, *p*: 0.060, respectively).

Positive associations were observed between acetic acid and propionic acid (r: 0.606, *p*: 0.008), butyric acid (r: 0.639, *p*: 0.004), and total SCFA (r: 0.928, *p*: 0.000). Propionic acid demonstrated a negative correlation with the quantity of Bifidobacterium and the relative abundance of other genera (r: −0.604, *p*: 0.004; r: −0.476 *p*: 0.046). Propionic acid also demonstrated a marginally positive association with the number of *L. plantarum* (r: 0.46, *p*: 0.052) and a marginally negative correlation with water consumption (r: −0.404, *p*: 0.097). Butyric acid had a minor association with protein consumption (r: 0.420, *p*: 0.083). The connection between total SCFA and the abundance of other genera was negative (r: −0.523, *p*: 0.026). Fecal pH correlated positively with Bacteroides (r: 0.527, *p*: 0.025) and marginally negatively with Agathobacter (r: −0.422, *p*: 0.081). Calories, carbohydrates, and protein intake had a negative correlation with Megamonas (r: −0.633, *p*: 0.005; r: −0.706, *p*: 0.001; and r: −0.548, *p*: 0.018, respectively). Meanwhile, fat had a slightly negative correlation with Agathobacter (r: −0.461, *p*: 0.054).

## 4. Discussion

The probiotic *L. plantarum* Dad-13 lowered significantly the HbA1c levels in the probiotic group in this study. Owing to concerns about COVID-19 transmission, not all responders were able to have their blood samples obtained. In the meantime, the HbA1c levels in the placebo group fell slightly but not significantly. However, the FBS and total cholesterol did not change significantly in the probiotic and placebo groups. This finding is consistent with a meta-analysis research which revealed that probiotics significantly reduced HbA1c levels but not FBS nor lipid profiles [26]. Another meta-analysis of several clinical studies revealed that probiotics may effectively lower fasting insulin, hemoglobin A1c, and FBG while enhancing the effectiveness of homeostatic model assessment of insulin resistance [6].

In this study, however, probiotic consumption did not alter the alpha (Figure 4) and beta diversities (Figure 5). The probiotics modulated some beneficial bacteria that are implicated in glucose metabolism. HbA1c had a negative connection with the quantitative numbers of *Bifidobacterium* (r: −0.669, *p*: 0.002) and the quantitative numbers of *Prevotella* (r: −0.566, *p*: 0.014) and relative abundances of *Succinivibrio* (r: −0.491, *p*: 0.038), *Faecalibacterium* (r: −0.658, *p*: 0.003), and *Collinsella* (r: −0.640, *p*: 0.004). HbA1c had a slightly positive association with body weight (r: 0.406, *p*: 0.095) and systolic blood pressure (r: 0.443, *p*: 0.066) but a slightly negative link with relative *Bifidobacterium* and *Bacteroides* abundance (r: −0.425, *p*: 0.079; r: −0.427, *p*: 0.077, respectively) (Figure 6).

*Bifidobacterium* has been linked negatively to T2D in the majority of studies [2,27]. The synthesis of acetate and lactate during carbohydrate fermentation, in which organic acids can be transformed into butyrate by other colon bacteria through cross-feeding interactions, is a significant *Bifidobacterium* function that supports gut homeostasis and host health [28]. This result is in line with that of another study that observed a reduction in *Succinivibrio* and *Faecalibacterium* among diabetic individuals compared with healthy controls [29]. Salamon [30] also observed that *Faecalibacterium* and *Collinsella* had a negative correlation with HbA1c. A prior study of Indonesian women showed that *Faecalibacterium* and *Prevotella* dominated in the non-T2D group [31]. *Prevotella* and *Succinivibrio* are high-polysaccharide-fermenting bacteria [32]. In five human case-control investigations, *Faecalibacterium prausnitzii* exhibited a negative association with T2D [27]. *Faecalibacterium prausnitzii* is one of the anaerobic bacteria that enables the frequent observation of butyrate in healthy human guts. By inhibiting histone deacetylase 1, which is specifically targeted by butyrate, the interleukin (IL)-6/signal transducer and activator of transcription 3/IL-17 pathway may be suppressed, which reduces inflammation and eventually enhances insulin sensitivity [26]. Meanwhile, *Bacteroides* is positively associated with the disease in some cases [33], whereas some reports suggest a negative association [3].

The F/B in the probiotic group increased, and that in the placebo group decreased after the intervention. Probiotics improved the dysbiosis in T2D patients because the F/B, which confirms dysbiosis in T2D individuals, decreased significantly. A prior study showed that T2D patients had a considerably greater F/B than non-T2D patients [30,31]. Previous research revealed that probiotics Lactobacilli and Bifidobacteria can specifically alter the intestinal microbiota (increase and reduce the levels of good and bad bacteria, respectively) and modulate metabolites, such as SCFA, trimethylamine N-oxide, bile acids, and indole propionic acids, which are linked to the regulation of glucose metabolism [34] and host immune response and play a beneficial role in the treatment of T2D [35].

The SCFAs are produced by intestinal gut bacteria via the fermentation of complex carbohydrates. These products may play a role in metabolic and energy homeostasis either directly or indirectly [36]. Numerous studies have shown that SCFAs influence gene expression, proliferation, and differentiation and act as substrates for gluconeogenesis and lipogenesis. The pharmacological effects of SCFAs on G protein-coupled receptors include the modulation of glucagon-like peptide 1, which is linked to improved insulin production and decreased blood glucose levels [36,37]. Acetic, propionic, and butyric acid concentrations and the total SCFA levels were not substantially different between the placebo and probiotic groups, but the probiotic group’s SCFA levels showed an increase.

After the intervention, neither group’s macronutrient, fiber, nor water intake changed significantly. In both groups, the only protein relative intake was in the range of World Health Organization (WHO) recommendations (10% to 15%). Meanwhile, the relative carbohydrate intake was lower (55% to 75%), and the fat relative ratio was higher than WHO recommendations (15–30%). Calories, carbohydrates, and protein intake had a negative correlation with *Megamonas* (r: −0.633, *p*: 0.005; r: −0.706, *p*: 0.001; and r: −0.548, *p*: 0.018, respectively). Meanwhile, fat had a slightly negative correlation with *Agathobacter* (r: −0.461, *p*: 0.054). *Megamonas* is a member of the Firmicutes and produces acetic, propionic, and lactic acids after fermenting different types of carbohydrates. *Megamonas* is important for obtaining organic nutrients. Healthy Yao nationals in China have low levels of *Megamonas* in their intestines, and this condition is connected to the unique healthy dietary practices of the ethnic group [38,39]. Gram-positive, anaerobic bacteria belonging to the *Lachnospiraceae* family are referred to as *Agathobacter* species. Butyrate, acetate, hydrogen, and lactate are the primary fermentation byproducts [40]. Healthy and high-fiber foods, such rice, cassava, tofu, and tempeh, were assessed as the main sources of calories, carbs, and protein in this study.

The consumption of probiotic powder *L. plantarum* Dad-13 did not show significant changes in weight, BMI, waist circumference, and hip circumference. This finding contrasts with that of a prior study, which found a significant decrease (*p* < 0.05) in the body weight and BMI of the overweight subjects after 90 days of *L. plantarum* Dad-13 ingestion, especially in the female subjects [13]. Compared with the placebo group, high consumption of *L. casei* for two months effectively reduced weight, BMI, and waist circumference in diabetic individuals [41]. However, a meta-analysis of 17 trials showed that probiotics did not change the BMI [42].

The systolic and diastolic blood pressures of T2D patients did not change after the consumption of the *L. plantarum* Dad-13 powder. The meta-analysis of 14 trials revealed that systolic and diastolic blood pressures decreased significantly [42]. However, the meta-analysis of five studies revealed that neither the usage of a single species probiotic nor the use of co-supplements resulted in a significant decrease in systolic blood pressure. A noticeable decrease in the systolic pressure was observed in multiple species of probiotics [35].

The *L. plantarum* Dad-13 treatment increased the probiotic group’s frequency of defecation. Based on the rise in *L. plantarum* concentration in the probiotic group, *L. plantarum* Dad-13 may survive in the digestive tract. However, no appreciable variations were noticed in the proportions of *Prevotella* and *Bifidobacterium* in either group. A limitation of this study was its small sample size. Confounding factors, such as the type and dosage of diabetes medication, might have also affected the results.

## 5. Conclusions

In conclusions, an 11-week intervention with *L. plantarum* Dad-13 powder improved the HbA1c level in the probiotic group. Probiotics may modulate some beneficial bacteria that promote metabolites, including SCFA alignment with the increment in total SCFA, acetic, propionic, and butyric acids.

## Figures and Tables

**Figure 1 microorganisms-10-01806-f001:**
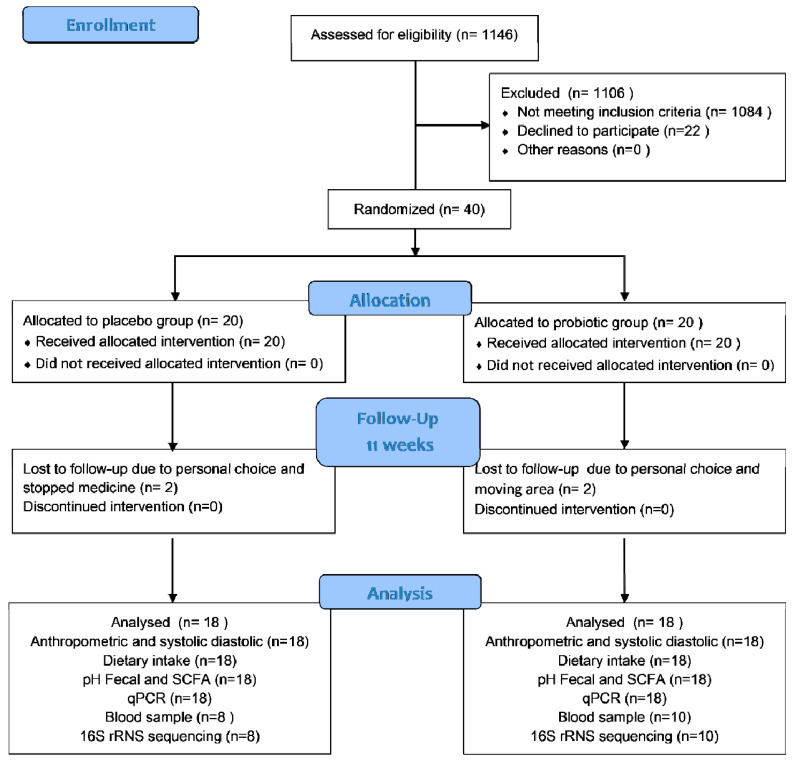
Participant flow according to CONSORT.

**Figure 2 microorganisms-10-01806-f002:**
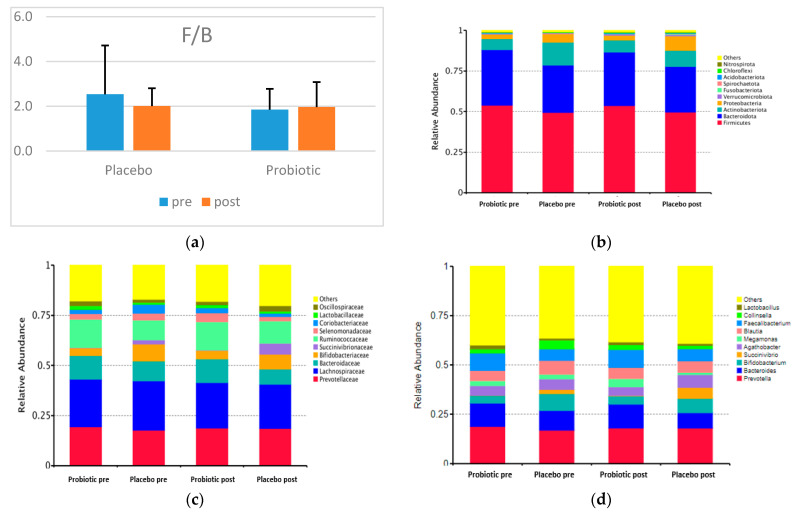
F/B and the top 10 relative abundances of GM composition between groups (**a**) F/B; (**b**) the top 10 relative abundances at the phylum level; (**c**) the top 10 relative abundances at the family level; (**d**) the top 10 relative abundances at the genus level.

**Figure 3 microorganisms-10-01806-f003:**
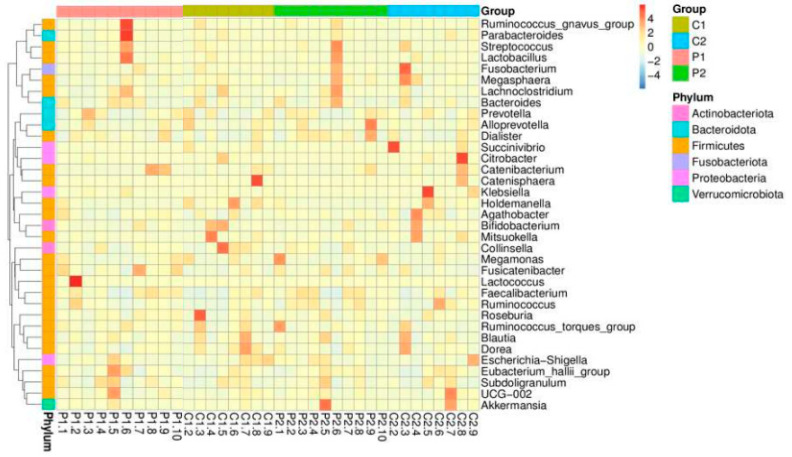
Heatmap of the top 35 relative abundances of each subject and group at the genus level.

**Figure 4 microorganisms-10-01806-f004:**
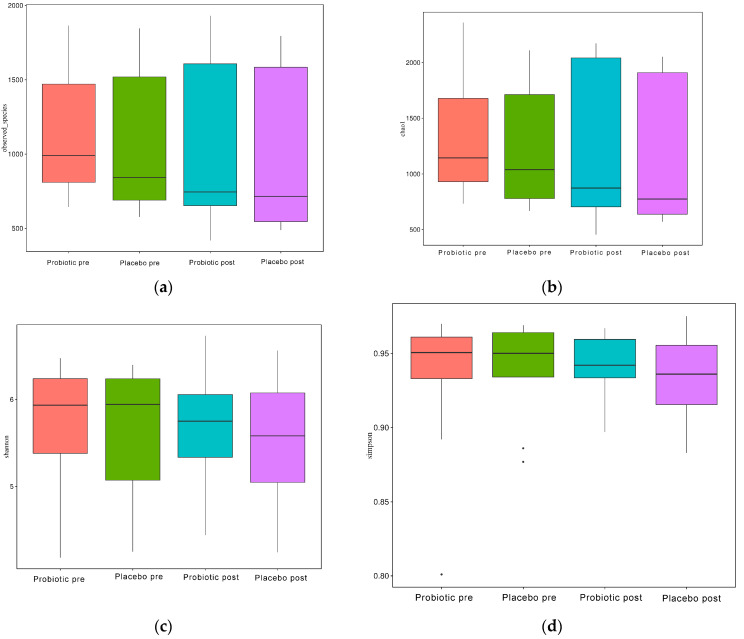
Alpha diversity boxplot index. (**a**) Observed species, (**b**) Chao1, (**c**) Shannon, and (**d**) Simpson.

**Figure 5 microorganisms-10-01806-f005:**
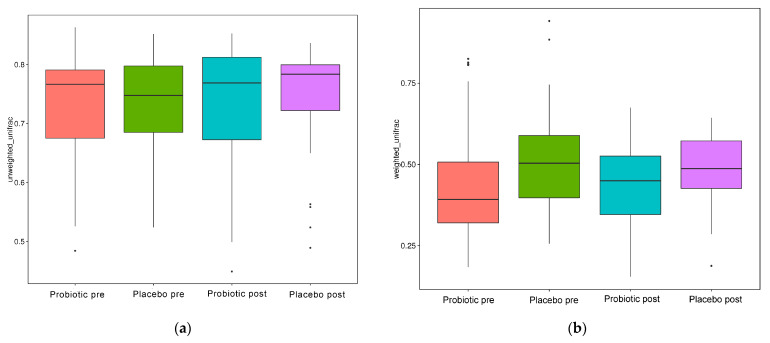
Beta diversity index box plot. (**a**) Unweighted and (**b**) weighted unifrac.

**Figure 6 microorganisms-10-01806-f006:**
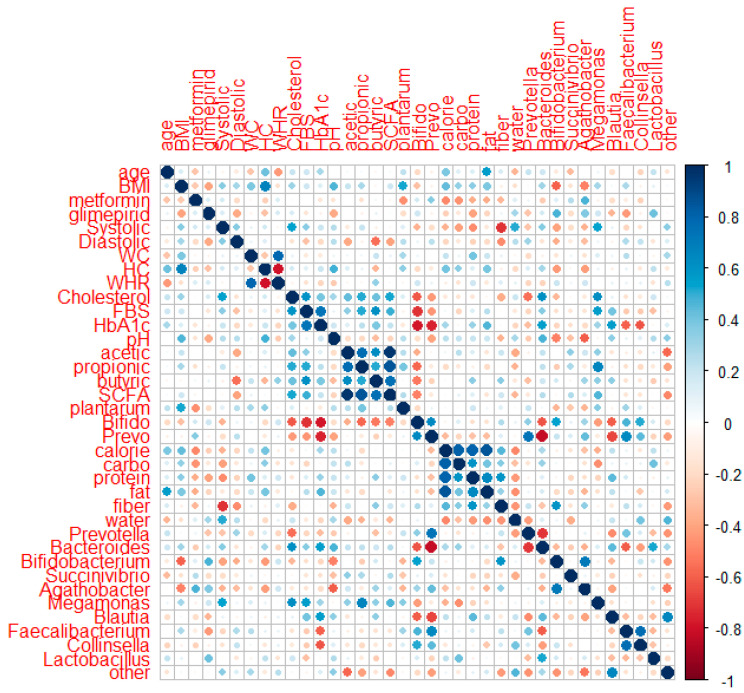
Corrplot of correlation between metabolic profiles, SCFA, GM at the genus level, and food intake in the placebo and probiotic groups after probiotic intervention.

**Table 1 microorganisms-10-01806-t001:** Specific primers used in the study.

Target	Primer	Sequence (5′ → 3′)	
*Lactobacillus plantarum*	sg-Lpla-F	CTC TGG TAT TGA TTG GTG CTT GCA T	[16]
	sg-Lpla-R	GTT CGC CAC TCA CTC AAA TGT AAA	
*Bifidobacterium*	g-Bifid-F	CTC CTG GAA ACG GGT GG	[17]
	g-Bifid-R	GGT GTT CTT CCC GAT ATC TAC A	
*Prevotella*	g-Prevo-F	CACRGTAAACGATGGATGCC	[17]
	g-Prevo-R	GGTCGGGTTGCAGACC	

**Table 2 microorganisms-10-01806-t002:** Characteristic of study subjects at baseline.

Characteristics	Placebo (n:18)	Probiotic (n:18)	*p*-Value
Age (year)	43.44 ± 4.44	44.11 ± 3.31	0.799
Weight kg)	57.11 ± 6.35	58.31 ± 7.94	0.620
Height (cm)	152.44 ± 4.66	150.90 ± 6.46	0.418
BMI (kg/m^2^)	24.58 ± 2.52	25.62 ± 3.15	0.279
Normal	10 (55.6%)	7 (38.9%)	
Overweight	8 (44.4%)	11 (61.1%)	
WC (cm)	85.64 ± 4.52	85.59 ± 8.17	0.980
HC (cm)	95.57 ± 4.78	95.97 ± 6.10	0.830
WHR	0.90 ± 0.04	0.89 ± 0.06	0.767
Systolic (mmHg)	126.56 ± 14.84	125.94 ± 15.32	0.904
Diastolic (mmHg)	83.06 ± 11.88	84.67 ± 9.04	0.650
FBS (mg/dL)	177.00 ± 77.22	184.22 ± 60.64	0.645
HbA1c (%)	9.36 ± 2.63	9.69 ± 2.08	0.358
Total cholesterol total (mg/dL)	195.11 ± 38.87	211.00 ± 52.99	0.312
Duration of T2D (years)	3.28 ± 2.14	3.32 ± 3.11	0.533
<1 year	0 (0%)	3 (16.7%)	
1–3 years	10 (55.6%)	9 (50%)	
3 years	8 (44.4%)	6 (33.3%)	
Antidiabetic drugs			
Met	7 (38.9%)	4 (22.2%)	
TZDs	-	2 (11.1%)	
Met + SU	10 (55.6%)	10 (55.6%)	
Met + TZDs	-	1 (5.6%)	
Met + SU + TZDs	1 (5.6%)	-	
SU + AGIs + TZDs	-	1 (5.6%)	
Dose/day			
Metformin			
500 mg	2	5	
1000 mg	10	9	
1500 mg	6	2	
Pioglitazone 30 mg	1	2	
Glimepiride			
0.5 mg	2	0	
1 mg	4	8	
2 mg	3	2	
3 mg	1	0	
4 mg	1	0	
Gliben 5 mg	0	1	
Gliabetes 30 mg	0	1	
Acarbose 100 mg	0	1	

BMI: body mass index; WC: waist circumference; HC: hip circumference; WHR: waist–hip ratio; Met: metformin; TZDs: thiazolidinedione; SU: sulfonylurea; AGIs: alfa glucosidase inhibitors.

**Table 3 microorganisms-10-01806-t003:** Anthropometric and blood pressure before and after intervention in both groups.

Parameters	Placebo (n = 18)		*p*	Probiotics (n = 18)		*p*
	Before	After		Before	After	
Weight (kg)	57.11 ± 6.35	57.22± 6.18	0.739	58.31 ± 7.94	58.26 ± 8.30	0.835
BMI (kg/m^2^)	24.58 ± 2.52	24.62 ± 2.50	0.738	25.62 ± 3.15	25.61 ± 3.33	0.876
WC (cm)	85.64 ± 4.52	84.89 ± 5.22	0.435	85.59 ± 8.17	85.29 ± 7.56	0.694
HC (cm)	95.57 ± 4.78	95.61 ± 5.64	0.939	95.97 ± 6.10	96.17 ± 5.42	0.829
WHR	0.90 ± 0.04	0.89 ± 0.04	0.531	0.89 ± 0.06	0.89 ± 0.06	0.702
Systolic (mmHg)	126.56 ± 14.84	123.28 ± 11.80	0.067	125.94 ± 15.32	121.22 ± 11.86	0.069
Diastolic (mmHg)	83.06 ± 11.88	80.50 ± 10.68	0.306	84.67 ± 9.04	84.67 ± 6.31	1.000

BMI: body mass index; WC: waist circumference; HC: hip circumference; WHR: waist–hip ratio.

**Table 4 microorganisms-10-01806-t004:** Fecal characteristics and defecation frequency.

Parameters	Placebo (n = 18)		*p*	Probiotics (n = 18)		*p*
	Before	After		Before	After	
Defecation frequency/7 day	5.78 ± 1.83	5.78 ± 1.63	1.000	6.06 ± 1.55	6.17 ± 1.86	0.026 *
pH fecal	6.21 ± 0.61	6.28 ± 0.39	0.371	6.28 ± 0.39	6.21 ± 0.37	0.369
Color of fecal						
Yellow	-	2 (11.1%)		-	-	
Yellowish-brown	6 (33.3%)	3 (16.7%)		5 (27.8%)	9 (50.0%)	
Brown	6 (33.3%)	5 (27.8%)		10 (55.6%)	4 (22.2%)	
Green	6 (33.3%)	8 (44.4%)		3 (16.7%)	5 (27.8%)	
Consistency						
Constipation	-	-		1 (5.6%)	-	
Normal	14 (77.8%)	15 (83.3%)		15 (83.3%)	18 (100%)	
Mild diarrhea	4 (22.2%)	3 (16.7%)		2 (11.1%)	-	

* *p* < 0.05.

**Table 5 microorganisms-10-01806-t005:** SCFAs of feces.

Parameters	Placebo (n = 18)		*p*	Probiotics (n = 18)		*p*
(mmol/g Feces)	Before	After		Before	After	
Total SCFA	24.02 ± 12.67	24.44 ± 9.64	0.907	21.68 ± 15.25	26.21 ± 13.30	0.467
Acetic acid	13.55 ± 5.95	14.65 ± 5.79	0.579	13.31 ± 8.65	15.94 ± 7.17	0.555
Propionic acid	5.96 ± 5.21	5.04 ± 2.72	0.879	4.25 ± 3.34	5.63 ± 3.54	0.658
Butyruc acid	3.10 ± 2.26	3.36 ± 1.92	0.811	3.03 ± 2.88	3.53 ± 2.05	0.112

The total SCFA consisted of the following acids: acetic, propionic, iso-butyric, butyric, iso-valeric, and iso-caproic acids.

**Table 6 microorganisms-10-01806-t006:** Dietary intake and physical activity before and after intervention between groups.

Parameter	Placebo (n:18)		*p*	Probiotics (n:18)		*p*
	Before	After		Before	After	
	Mean ± SD (%Energy)	Mean ± SD (%Energy)		Mean ± SD (%Energy)	Mean ± SD (%Energy)	
Energy (Kcal)	1570.69 ± 479.01	1569 ± 453.23	0.952	1481.74 ± 378.27	1463.71 ± 368.45	0.435
Carbohydrate (g)	175.28 ± 49.61 (45.8)	182.37 ± 35.22 (48.6)	0.286	185.39 ± 54.12 (49.9)	180.04 ± 56.09 (48.9)	0.456
Protein (g)	53.15 ± 18.66 (13.5)	52.26 ± 21.55 (13.0)	0.686	48.36 ± 15.33 (13.2)	48.81 ± 15.73 (13.3)	0.833
Fat (g)	76.78 ± 33.48 (42.6)	73.02 ± 33.16 (39.9)	0.302	63.50 ± 19.43 (38.6)	63.96 ± 17.77 (39.7)	0.891
Fiber (g)	11.67 ± 6.15	12.08 ± 4.42	0.669	11.18 ± 4.85	10.28 ± 2.28	0.324
Water (g)	1859.53 ± 806.90	1888.32 ± 743.20	0.701	2028.28 ± 548.31	1977.03 ± 632.33	0.468
Physical activity (MET)	9099.00 ± 3696.38	7886.64 ± 2869.80	0.099	10,218.53 ± 3433.14	9152.67 ± 3337.44	0.230

**Table 7 microorganisms-10-01806-t007:** Specific bacteria analyzed by PCR.

Parameters	Placebo (n:18)		*p*	Probiotics (n:18)		*p*
Log 10 Bacterial Cell/g Feces	Before	After		Before	After	
*L. plantarum*	4.46 ± 0.47	4.15 ± 0.36	0.020 *	4.72 ± 0.49	5.56 ± 0.63	0.001 *
*Bifidobacterium*	7.16 ± 0.76	7.16 ± 0.75	0.811	7.02 ± 0.59	7.15 ± 0.89	0.185
*Prevotella*	7.55 ± 1.02	7.69 ± 0.75	0.170	7.35 ± 1.10	7.29 ± 1.14	0.798

* *p* < 0.05.

**Table 8 microorganisms-10-01806-t008:** Metabolic profiles before and after intervention in both groups.

Parameters	Group	Before	After	*p*	Change	*p*
FBS (mg/dL)	Placebo (n:8)	170.75 ± 85.62	180.25 ± 73.43	0.575	9.50 ± 38.35	0.393
	Probiotic (n:10)	175.80 ± 63.55	164.50 ± 70.04	0.678	−11.30 ± 57.36	
	*p*	0.964	0.214			
HbA1c (%)	Placebo (n:8)	9.28 ± 2.70	8.80 ± 2.58	0.103	−0.48 ± 0.72	0.533
	Probiotic (n:10)	9.14 ± 2.29	8.04 ± 2.01	0.008 *	−1.10 ± 1.78	
	*p*	0.824	0.476			
Total Cholesterol (mg/dL)	Placebo (n:8)	182.63 ± 33.47	189.38 ± 24.63	0.274	6.75 ± 16.08	0.096
	Probiotic (n:10)	210.20 ± 52.74	203.40 ± 47.48	0.218	−6.80 ± 16.25	
	*p*	0.218	0.461			

* *p* < 0.05; FBS: Fasting blood sugar.

## Data Availability

The data presented in this study are available on request from the corresponding author. The data are not publicly available due to privacy protection.

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
