# Peer review of "Effect of Probiotic Lactobacillus plantarum Dad-13 on Metabolic Profiles and Gut Microbiota in Type 2 Diabetic Women: A Randomized Double-Blind Controlled Trial"

_microorganisms, 2022, doi:10.3390/microorganisms10091806_

Round 1

Reviewer 1 Report

The study carried out by the Ninik Rustanti et.al. concerned the assessment of the effect of Lactobacillus plantarum Dad-13 on selected metabolic parameters and the microbiota of the gastrointestinal tract in patients with type 2 diabetes. In research used i.a. molecular techniques: DNA isolation, real-time PCR amplification and sequencing.

Regarding real-time PCR, there is no information whether standard curves were made based on the reference strains Bifidobacterium and Prevotella and Lactobacillus plantarum.

In the context of bacterial quantification by real-time PCR, have you considered the use of species-specific probes? Reactions with the use of the Eva Green dye, despite the primers used, may cause the amplification of non-specific products, which may translate into overestimation of the quantitative results.

Moreover, the obtained quantitative results on Lactobacillus plantarum, Bifidobacterium and Prevotella should be correlated with data on metabolic parameters (including HbA1c). The quantitative results will be more accurate than those obtained from sequencing which allows a semi-quantitative evaluation. Perhaps then it will be possible to obtain a stronger correlation.

After electrophoretic separation only one band was obtained per sample? If yes, there is no need to cut and isolate the DNA from electrophoretic gel because it  contributes to the loss of material to sequencing

In the results section, I propose to limit the number of tables, and transfer some of them to a supplement (eg Table 4, 5 and 6). In the study a lot of different analyses were carried out, and the number of data, which is often statistically insignificant, is very large. After analyzing them, it is difficult to catch the most important information.

Table 8 lacks an asterisk at p = 0.008

Are the changes in the decrease or increase of genera such as Prevotella, Faecalibacterium, Bifidobacterium etc. were statistically significant? If not, this information should be included in the description.

It is also suggested to standardize the abbreviation type 2 diabetes mellitus - T2DM or T2D throughout the manuscript, and to replace the word flora with microbiota (line 391).

Reviewer 2 Report

Reviewer comments and suggestions

This study of this study was to assess the effect of the probiotic L. plantarum Dad-13 powder on metabolic profiles and gut microbiota (GM) of women with T2D in Yogyakarta Indonesia. For this study, the authors recruited twenty women from each group of forty T2D patients who received either a probiotic or a placebo. The probiotic group consumed 1 g skim milk powder containing 1010 CFU/g L. plantarum daily for 11 weeks. The placebo group received 1 g skim milk powder only daily for 11 weeks. 

The study result included HbA1c in the probiotic group (n: 10) which showed a significant decrease from 9.34% ± 2.79% to 8.32% ± 2.04%. However, in comparison with the placebo (n: 8), L. plantarum Dad- 13 supplementations did not significantly decrease the HbA1c level. Additionally, the study suggested that the GM analysis resulted into L. plantarum Dad-13 supplementation caused a considerable increase in the L. plantarum number. Moreover, no significant change was observed in the fecal pH and SCFA after supplementation with L. plantarum Dad-13.

Overall the manuscript is average for publication. The authors need to proofread the manuscript again so that typo errors could be minimized. A few concerns are below to be incorporated in the revised version of the manuscript.

  1. Somewhere, diabetes is written as a disorder and somewhere as a disease. Please be consistent 
  2. In the abstract, lines 20-21, the sentence this probiotic….. needs to be corrected.
  3. In the whole paper somewhere type 2 diabetes mellitus has mentioned as T2D and somewhere as T2DM.
  4. In lines 50 and 55, the grammar needs to be checked for these lines.
  5. In lines 67-69, the sentence is not clear.
  6. In line 24, the dose is mentioned as skimmed milk with 1 g L. plantarum while in line 80, only 1 g L. plantarum has written and the CFU value has written in the wrong manner.
  7. In lines 98-99, the sentence has repeated.
  8. Comments for table 6 Did the authors discuss these precautionary in the material and method section
  9. In section 2.6, all the reagents have taken in such a huge amount i.e. in hundreds of liters. Please check 
  10. In section 3.10, lines 349-353, the meaning of both sentences is almost the same.
  11. Discussion: First Para needed to be discussed with authors own result. No need to extend the same information
  12. Line 409-411 Please mention the figure or table used for these results
  13. Line 421 Salamon (34) what does it represent, please check and correct it.
  14. In line 398, mistakenly HbA1 has written.
  15. Line 437 as the authors mention more research revealed but did not cite references in a proper place, better to change the form of the sentence.
  16. In line 444, the spelling of homeostasis is wrong.
  17. Line 468, the authors report no significant changes, it would be nice to discuss the reason for this
  18. The authors have to modify all the references based on MDPI guidelines.
  19. In reference, line 524, the spelling of diabetes is wrong.
  20. Related to the references: square bracket, not round bracket, please modify in the manuscript. 
  21. In references, lines 534, 571, and 590, the page numbers are not written properly.
